# Life-Cycle Modeling of Structural Defects via Computational Geometry and Time-Series Forecasting [note 1]

**DOI:** 10.3390/s19204571

**Published:** 2019-10-21

**Authors:** Sara Mohamadi, David Lattanzi

**Affiliations:** Department of Civil, Environmental, and Infrastructure Engineering, George Mason University, Fairfax, VA 22030, USA; smohama2@gmu.edu

**Keywords:** remote sensing, photogrammetry, life-cycle modeling, time-series forecasting, structural damage, stochastic modeling, convex hull, ARIMA, VAR, fatigue crack prediction

## Abstract

The evaluation of geometric defects is necessary in order to maintain the integrity of structures over time. These assessments are designed to detect damages of structures and ideally help inspectors to estimate the remaining life of structures. Current methodologies for monitoring structural systems, while providing useful information about the current state of a structure, are limited in the monitoring of defects over time and in linking them to predictive simulation. This paper presents a new approach to the predictive modeling of geometric defects. A combination of segments from point clouds are parametrized using the convex hull algorithm to extract features from detected defects, and a stochastic dynamic model is then adapted to these features to model the evolution of the hull over time. Describing a defect in terms of its parameterized hull enables consistent temporal tracking for predictive purposes, while implicitly reducing data dimensionality and complexity as well. In this study, two-dimensional (2D) point clouds analogous to information derived from point clouds were firstly generated over simulated life cycles. The evolutions of point cloud hull parameterizations were modeled as stochastic dynamical processes via autoregressive integrated moving average (ARIMA) and vectorized autoregression (VAR) and compared against ground truth. The results indicate that this convex hull approach provides consistent and accurate representations of defect evolution across a range of defect topologies and is reasonably robust to noisy measurements; however, assumptions regarding the underlying dynamical process play a significant the role in predictive accuracy. The results were then validated on experimental data from fatigue testing with high accuracy. Longer term, the results of this work will support finite element model updating for predictive analysis of structural capacity.

## 1. Introduction

This paper presents a method for modeling the time-history evolution of defects quantified through remote sensing technologies such as laser scanning, photogrammetry, or digital imaging. The goal is to expand the use of structural assessment information commonly collected during routine inspections and improve the structural life-cycle assessment process. Conventional structural assessments are typically based on visual inspections, embedded sensor systems, nondestructive evaluation (NDE) techniques, or some combination thereof [1,2,3,4,5,6,7]. In almost all cases, the goal is the nondestructive detection and identification of structural performance changes and damage, as well as to assess the reliability and safety of monitored structures [8,9]. In addition to immediate structural evaluation, these assessments ideally help engineers to estimate the remaining life of structures. This is commonly done by reviewing historical performance records and holistically identifying temporal trends from the assessment data. However, most assessment data are not structured in a way that explicitly captures the life-cycle performance of a structure, and it is, therefore, challenging to quantitatively evaluate the evolution of inspection data over time and carry out a predictive analysis of the future state of the structure. Life-cycle data modeling advancements could provide more precise and robust structural information, leading to better system asset management decision-making, with apparent safety and financial benefits.

### 1.1. Prior Work

Damage identification, localization, and quantification were extensively studied in the last few decades. Using state-of-the-art NDE methods, cracks can be identified and located using vibration-based methods [10,11,12]. These damage identification techniques can be categorized into four main categories based on vibration characteristics: natural frequency-based, mode shape-based, mode shape curvature-based, and techniques using both mode shapes and frequencies. Fan et al. [10] presented a comprehensive review on these parameter-based methods and discussed their advantages and drawbacks. They also conducted a comparative study of five widely used damage detection algorithms for beam-type structures to assess the validity and effectiveness of the signal processing algorithms. While effective, these methods only focus on damage identification and localization, whereas the development of quantification techniques for damage magnitude is limited. Large cracks and voids in concrete, as well as corrosion and cracks in steel, can be identified and located using ultrasonic tests [13,14,15]. Sharma et al. [13] used ultrasonic guided waves to monitor beams undergoing accelerated impressed current corrosion in the presence of two corrosion mechanism (chloride and oxide). It was found that ultrasonics can successfully detect corrosion and identify the specific corrosion mechanism. Rens et al. [15] presented the concept and application of a new indirect inspection technique using ultrasonic spread-spectrum methods to test structural objects. Their laboratory findings show that this new method may be feasible for monitoring and evaluating existing large or complicated structural members. Acoustic wave techniques can also be used to identify and locate imperfections such as the initiation of crack and the growth rate of fatigue cracks and corrosion [16], to classify crack modes in concrete [17,18,19], and to quantify the severity of damage [20,21]. Ohno et al. [18] conducted two crack classification methods and showed that tensile and shear cracks can be distinguished using their criteria. Sagar et al. [16] used acoustic emission parameterizations to classify the severity of damage into minor, intermediate, and severe damage categories. These prior works show that the acoustic emission techniques are often capable of detecting damage in their early stages, so that an early warning can be given to allow for repair work before a structural element is seriously damaged. Furthermore, it is possible to detect and localize hidden and subsurface defects using radiographic tests [22,23] and electromechanical methods such as ground penetration radar [24]. Chen et al. [24] used ground-coupled penetrating radar (GCPR) to characterize the subsurface conditions of three roadway pavements. The extents of the anomalies in the horizontal and vertical direction were visible in GCPR images, and this study successfully demonstrated that GCPR is able to identify anomalies and voids.

While these prior studies created the capacity to identify and quantify damage to structures, the emphasis was on improving the accuracy of the detection and measurement of defects. To date, significant efforts were not made to model these measured changes/defects as time-dependent phenomena. Advancements in understanding and modeling of temporal defects will lead to improved decision-making capabilities, as well as expanded use of sensing and monitoring technologies.

In addition to the development of NDE and structural health monitoring (SHM) techniques, there are increasing studies on the use of remote sensing and imaging technologies such as LiDAR or photogrammetry to provide new sources of inspection information. These remote sensing technologies provide high-resolution two-dimensional (2D) images or three-dimensional (3D) point cloud models of structures, and can capture the small-scale defects that are critical to understanding structural performance [25,26,27,28,29]. In complement to the expanding use of these technologies, there are now a variety of methods for isolating and extracting defects from 2D or 3D images [30,31], and advancements in deep machine learning methods portend future improvements [32,33]. A key advantage of these data sources is the direct link between quantified geometric changes and changes in the underlying mechanical performance that can be captured in finite element analysis, as evidenced by a variety of prior work [34,35,36]. While such capabilities provide valuable tools for structural assessment, they do not explicitly quantify life-cycle dynamics and forecasts of future defect conditions.

The question of the reliability of an engineered system led researchers to investigate the growth of defects such as fatigue cracks and corrosion over the life cycle of the systems. To study fatigue crack growth, model-based estimation methods such as Bayesian methods [37], extended Kalman filtering [38], and Monte Carlo sampling [39] were used for quantification of the estimation uncertainty. For corrosion, theoretical models and simulation tools were developed for a better understanding of the nature of the pitting corrosion process, to allow prediction of the temporal evolution of maximum pit depth in corroding structures. In recent studies, stochastic approaches were also applied to simulate corrosion [40,41,42,43]. All these efforts primarily focus on estimating the reliability of a system given estimates of the current state of a defect, rather than quantifying the future state of the defect in a way that provides support for predictive capacity assessments.

### 1.2. Contribution of This Research

As stated, a critical aspect of long-term structural monitoring is an understanding of detected defects as time-dependent phenomena, and this remains a major outstanding research need. Enhanced temporal modeling of defects would lead to the ability to predict future defect states and, thus, predict the future condition of the structure. While prior studies investigated the temporal behavior of defects from an empirical perspective [44,45], efforts to quantify the dynamics of defect observations captured in remote sensing were, to date, limited. The main objective of this study is to address these limitations for life-cycle modeling of remotely sensed defects using a computational geometry approach to defect parametrization combined with time-series modeling.

Presented in this paper is a novel algorithm to model the life cycle of defects manifested as either 2D or 3D point clouds. Point cloud data are geometric representations of defects that can serve as a consistent record of the state of a structure at a given inspection interval and provide a basis for finite element analysis, among other uses [25,34]. This algorithm extracts latent features from these defect point clouds through computational geometry, fits a time-series process model to the evolution of those features, and uses a stochastic forecasting model to predict the future state of the defect. Consequently, a predictive analysis with regard to the future condition of the structure could be carried out by linking the modeled evolution of defects to a numerical simulation, ultimately helping to provide a complete representation of structural performance and integrity over time. This paper does not consider the predictive simulation aspect of this process, and the readers are referred to References [34,35,36,46] for potential applications in this domain.

The remainder of this paper is structured as follows: firstly, the complete analytical methodology is presented. This is followed by a presentation of synthetic experimental results designed to illustrate the key behavioral aspects of the algorithm. This is followed by experimental evaluation using fatigue crack propagation data. The paper concludes with an overall assessment of the algorithm and avenues for future work.

## 2. Materials and Methods

The methodology presented here consists of several steps (Figure 1). Once a defect in the structure is detected, the defect must be parametrized in a way that can support a dynamic modeling of defect evolution over time. Parameterization is achieved with feature extraction from the point cloud through computational geometric modeling of the convex hull of the cloud, resulting in a combination of hull simplexes and vertices. These parameterizations are computed for multiple time steps over the life cycle of the structure. Once parameterized, a time-series model is fit to the sequence of parameterizations in order to capture the underlying process of evolution. This model fitting also enables the estimate the future state of the defect via out-sample forecasting. The defect shape is then reconstructed by reversing the parameterizations back to a geometric point set. While not considered in this work, the methodology presented here can then be used to update a finite element model of the structural component based on the predicted future defect topology, leading to predictive structural assessment.

### 2.1. Defect Parameterization

Once a remotely sensed defect in a structural component is detected through computer vision [34], the first step in the modeling process is to parameterize it so that a stochastic dynamic model can be reliably fit the extracted parameters, or a “feature vector” to track the defect evolution over time. The complex nature of point cloud data necessitates this low-dimensional parameterization, as tracking each individual point in a cloud would lead to an intractably high number of time-series model coefficients.

Here, we propose that the feature extraction can be done using the concept of a geometric convex hull. The convex hull of a point set is a unique representation of a point set in *R^n^*, defined as the smallest convex polygon that surrounds all points in the point set (Figure 2) [47]. In *R*^3^ or higher-dimensional data spaces, the convex hull is similarly defined as the minimum convex polyhedron of the point set. It should be noted that, while this parameterization reduces the information complexity of the point cloud, the use of a convex hull serves to maintain the dimensionality of the underlying geometric object under observation. This is critical for reconstructing predicted defect states.

The determination of the convex hull is a geometric computation that is useful for many analyses and was successfully applied in domains such as image processing [48] and pattern recognition [49]. Although a number of alternative feature extraction approaches were considered in this study [50,51], hull parameterization was used because of its inherent advantages. The description of a defect in terms of its parameterized hull allows for consistent temporal tracking for predictive purposes, while also reducing the dimension and complexity of the data implicitly. In addition, the convex hull concept can be extended to high-dimensional spaces to support the fusion of multiple sensors and data types, a longer-term goal of this work.

The convex hull of point cloud *P* is a uniquely defined convex polygon. A natural way to represent a generalized polygon is by listing its vertices in clockwise order, starting with an arbitrary starting point. As such, the problem to be solved is as follows: given a point set *P* = {*p*_1_, *p*_2_, …, *p_m_*} in *R^n^*, compute a list that contains those points from *P* that are the vertices of the convex hull, CH(*P*). To find those vertices, the algorithm sorts all points through a “divide and conquer” approach [52]. The convex hull algorithm finds two points with maximum and minimum spatial coordinates in a single dimension and computes a line joining these two points. This line divides the whole set into two halves. For a given half it finds the points with a maximum distance from the dividing line, forming a triangle defined by minimum and maximum point distances. Those points inside the triangle are determined to not be part of convex hull. Then, these steps are iteratively repeated to search for points with maximum distance from the separating line, until there is no point left outside of the computed triangles. The points selected at this step constitutes the convex hull. The convex hull results in a vector containing the Cartesian coordinates of the hull vertices. This extracted vector of vertices is the numerical descriptor that is later fit to a stochastic process model (Figure 3).

### 2.2. Hull Registration and Structuring for Time-Series Modeling

Once the hull vertices are extracted, the data must be structured and compiled for time history modeling. Firstly, point clouds from each time step are aligned and registered in a common reference frame. In this work, manual registration was sufficient; however, automatic registration approaches such as the iterative closest point (ICP) algorithm, or by determining the 2D or 3D homography between point sets via feature-based computer vision methods, may be necessary (Figure 4) [53,54]. The registered defect point clouds are then parameterized using the convex hull computation, resulting in individual 2 × *n* arrays of vertex coordinates. Each vertex at *t* = *t*_1_ is then spatially tracked across the temporal sequence of vertex arrays. The vertices of a hull in stage 1 (hull_1_) are matched to their nearest neighbor (NN) vertices in *hull*_2_ and, likewise, those are matched to their NN vertices in hull_3_, and so on [55]. The assumed movement of one of the vertices is shown in Figure 5. The nearest neighbor is found based on the smallest Euclidean distance between vertex sets. Equation (1) shows the nearest neighbor search in $Hull(Q):\left\{q_{1},q_{2},\ldots ,q_{n}\right\}$ from *p*_1_ ∊ $Hull(P):\left\{p_{1},p_{2},\ldots ,p_{m}\right\}$ in 2D space (Hull(Q) and Hull(P) are two consecutive descriptor vectors).
(1)$$d(p_{1},q_{i})=\sqrt{{\left(q_{ix}-p_{1x}\right)}^{2}+{\left(q_{iy}-p_{1y}\right)}^{2}}.$$

At each time step, the NN of vertices in two time steps are determined, and the change in magnitude and orientation between matched vertices is computed and compiled for the complete temporal sequence (Figure 5). This resulting dataset consists of the distances and orientations of vertex changes between subsequent hull stages throughout a time series. This dataset is illustrated for the identified and aligned multiple stage of a defect with a polygon shape in Figure 4 and Figure 5.

### 2.3. Time-Series Modeling

The overall approach to time-series modeling is illustrated in Figure 6 and presented here. Once the dataset representing the evolution of defects is constructed (Section 2.2), a stochastic model can be fit to the dataset to model the dynamics of defect evolution. Each column of this dataset is a time series that represents the evolution of each feature over time. Time-series forecasting is performed in order to capture the underlying long-term life-cycle trends in inspection data. The model can then be used to extrapolate the time series into the future. This modeling approach is particularly useful when the temporal behavior is stochastic, as opposed to understood deterministic evolution, and where the relationship between parameterization variables is not well understood [56]. For example, in the problem presented later in this paper, there is no knowledge available on the boundary conditions (i.e., applied load and support reactions) of the tested structural component, and there is no established deterministic model for the propagation of fatigue cracks.

There are several common approaches to time-series modeling, including autoregression, moving average, exponential smoothing, autoregressive integrated moving average (ARIMA), and multivariate time-series vectorized autoregression (VAR). All of these models are linear, meaning that their predictions of the future values are constrained to be linear functions of past observations. Because of their effectiveness and ease of implementation, linear models are the main research focus for time-series modelers, as is the case in this study. Nonlinear modeling approaches such as recurrent neural networks were not considered here due to the limited training data available for observing defect evolution.

From the available time-series models, ARIMA and VAR models were selected for this study, and their performances were compared against each other. The autoregressive integrated moving average model (ARIMA) methodology developed by Box and Jenkins [57] is able to handle non-stationary time series, in other words, scenarios where the statistical properties of the time series measurements do not remain constant over time. As such, it relaxes the requirement that time-series data be covariance-stationary prior to modeling, and it is well suited to the challenging variations in field conditions that impact remote sensing-based inspection practices [58]. With ARIMA, the future value of a variable is assumed to be a linear function of several past observations and random errors. An ARIMA model contains two sub-processes (autoregressive, moving average process) and explicitly includes differencing in the formulation to account for the non-stationarity of the data. ARIMA gained enormous popularity in many areas, and research practice confirmed its power and flexibility [59,60,61]. A generalized form of the ARIMA time-series model is shown in Equation (2).
(2)$$Y_{\left(t\right)}=\phi_{0}+\phi_{1}\times Y_{\left(t-1\right)}+\phi_{2}\times Y_{\left(t-2\right)}+\ldots +\phi_{p}\times Y_{\left(t-p\right)}+\varepsilon_{t}+\theta_{1}\times \varepsilon_{\left(t-1\right)}-\theta_{2}\times \varepsilon_{\left(t-2\right)}-\ldots -\theta_{q}\times \varepsilon_{\left(t-q\right)},$$
where Y(t) and $\varepsilon_{t}$ are the actual value and random error at time *t*, respectively. $\phi_{i}$ (*i* = 1, 2, …, *p*) and $\theta_{j}$ (*j* = 0, 1, 2, …, *q*) are the vectorized model coefficients. *p* and *q* are integers, often referred to as orders of the model.

As an alternative, vector autoregressive (VAR) models are effective for multivariant time-series data where there is a potential dependency between model variables. Therefore, unlike an ARIMA model that estimates the present value of a variable based only on its past values, VAR models consider past values of other variables as well. This model was used here to study the dependency and effect of the movement of convex hull vertices on other vertices. The VAR model [62] was used for damage detection studies previously [63,64], but there was no effort to model the life cycle of a defect to date. The general form of the autoregressive model is shown in Equation (3). Similar to ARIMA, the VAR model relates the current value of a variable to its past values.
(3)$$Y_{\left(t\right)}=\phi_{0}+\phi_{1}\times Y_{\left(t-1\right)}+\phi_{2}\times Y_{\left(t-2\right)}+\ldots +\varepsilon_{t},$$
where $\varepsilon_{t}$ is random error (random shock) and $\phi_{i}$ represents a constant. Similarly, the VAR model relates the current value of a vector to its past values, and each variable depends not only on its own past values but also on those of other variables, in which *K* × 1 is a random vector, $\phi_{i}$ represents a fixed (*K* × *K*) coefficient matrix, and $\varepsilon_{t}=(\varepsilon_{1t},\ldots ,\varepsilon_{kt})$ is a *K*-dimensional random error [62]. To summarize, the power of ARIMA modeling stems from its ability to handle non-stationary time series with non-constant statistical properties over time. On the other hand, VAR is suitable for analyzing stationary multivariant time series with constant statistical properties and dependancy among variables.

#### 2.3.1. Model Parameter Identification

Prior to fitting the coefficients of the time-series model, the model order must be optimized. The ARIMA model contains three components for which an order must be determined: autoregressive (AR), integrated (I), and moving average (MA) (Equation (2)). The AR component uses the dependent relationship between an observation and some number of lagged observations. The order of the AR component (*p*) is the number of lag observation included in the model. The integrated component (I) employs differencing of raw observation data in order to make the time series stationary, and its order (*d*) is the number of times that the raw observation is differenced. The MA component uses the dependency between an observation and a residual error, and its order (*q*) is the size of the moving average window. In this study, ARIMA model orders (*p*, *d*, and *q*) were evaluated based on a mean squared error. A prototyping dataset was divided into train and test sets, and the optimized combination of model orders was chosen such that they produced the least mean squared error in the test set. For the VAR model, the model order, *p*, was selected based on Hannan–Quinn information criterion (HQIC) [65]. This criterion was applied because this is used to consistently estimate the order under fairly general conditions [66]. The criterion is shown in Equation (4).
(4)$$HQIC=-2\ln(L_{\max})+2k\ln(\ln(n)),$$
where *n* is the number of observations, *k* is the number of parameters to be estimated (e.g., the normal distribution has mu and sigma), and *L_max_* is the maximized value of the log-likelihood for the estimated model. The coefficients for *k* indicate the level to which the number of model parameters is being penalized. The objective is to find the model order of the selected information criterion with the lowest value HQIC value.

#### 2.3.2. Forecasting and Defect Reconstruction

Once a model is fitted to a sequence of defect observations, the future state of the convex hull parameterizations can be predicted by the forecasting model. Once predicted, the future defect shape can then be reconstructed by converting the feature vector into a hull shape. A complication is that the number of extracted features may be inconsistent at different time steps, and this discrepancy in the length of the vectors in some cases leads to an inaccurate defect prediction. This case will happen when the number of features extracted by convex hull computation at early time steps is significantly smaller than that in the later time steps. To handle this issue, a statistical assumption is employed. For features that were not fit to a model and, therefore, their values were not predicted by dynamic modeling, the arithmetic mean of other features can be used as their expected value.

The pseudocode for the complete algorithmic methodology is shown in Figure 7. Upon reconstruction of the predicted geometric configuration of a defect, it is then possible to update a numerical simulation to account for the predicted change in the structure’s geometry due to the defect. This updating process is not tested here, but such capabilities were developed in prior related work, including efforts by the authors [34,35,36].

## 3. Experimental Validation

This section presents and discusses the results of experiments designed to evaluate the developed methodology. Two series of tests are presented. The first tests involve a set of experiments performed on synthetic datasets. These datasets were designed to highlight key aspects of the modeling approach and provide insight into algorithm behaviors. The second tests are derived from laboratory-scale tests of fatigue crack propagation in aluminum tensile specimens, in order to illustrate the behavior of the modeling approach in a realistic use case.

### 3.1. Synthetic Dataset

To initially test the accuracy and robustness of the presented methodology, synthetic 2D point clouds analogous to data derived from remote sensing (e.g., laser scanning or photogrammetry) were generated over simulated defect life cycles. Synthetic point clouds with distribution characteristics representing different flaw topologies (e.g., rectangle, circle, and generalized polygon) were generated. Additionally, varying evolution time histories were generated synthetically, representing a variety of stochastic processes (e.g., linear, quadratic, random Gaussian, and random uniform). For every combination of defect shape and stochastic process, a set of 20 defect time steps was generated. White noise was also introduced into the point clouds for each time step, in order to simulate more realistic measurements. Finally, uniform and non-uniform feature evolution were considered. Uniform feature evolution refers to a case where all vertices of the convex hull (features in the extracted descriptor vector) have the same expansion magnitude regardless of the trend of evolution. Cases where vertices were allowed to expand at varying magnitudes over a life-cycle simulation were considered non-uniform.

#### 3.1.1. Time-Series Stationarity Assessment

Time series are stationary if the statistics calculated on the time series (e.g., the mean or variance of the observations) are consistent over time. Most statistical modeling methods assume, or require, the time series to be stationary to be effective. There are many methods to check whether a time series is stationary or non-stationary, such as reviewing a time-series plot, reviewing the summary statistics for time series, or using statistical tests. The augmented Dickey–Fuller test [67], one of the more widely used, was used in this study. It uses an autoregressive model and optimizes an information criterion across multiple different lag values. The null hypothesis (H0) of the test is that the time series can be represented by a unit root that it is not stationary (i.e., it has some time-dependent structure). The alternate hypothesis (rejecting the null hypothesis) is that the time series is stationary.

Table 1 shows the average *p*-values of each generated time series, which is used in the augmented Dickey–Fuller test to evaluate stationarity for various defect shapes and evolutions. A *p*-value above 0.05 suggests that a test fails to reject the null hypothesis (H0), and it is concluded that such time-series models are non-stationary. This analysis shows that all of the generated time-series simulations, with the exception of the simplest linear evolution process, are non-stationary. As such, it was anticipated that the ARIMA approach would perform better than the VAR approach for the synthetic datasets.

#### 3.1.2. Time-Series Modeling

After the synthetic datasets were generated, ARIMA and VAR model orders were selected prior to fitting to time series. For the VAR models, all extracted feature vectors were input as a matrix at once, and the model order (*p*) was set as the same for all features, whereas, for the ARIMA model, each feature vector was considered individually, and model orders suitable to each time series were chosen based on least mean squared error. Models were then fit to the time series.

#### 3.1.3. Metrics

The key metric for evaluating the time-series model behavior was defect reconstruction accuracy. To evaluate the results, the predicted defect shape for a future time step was compared against an established ground truth for each scenario. Two geometric metrics were computed: the percentage difference between the area of two shapes and their overlap area percentage. These two metrics were necessary in order to identify scenarios where the predicted and ground truth defect sizes were similar, but where there was a divergence in the geometric topology. Measuring point clouds directly on a point-wise basis using the raw data was not considered, as the randomly sampled nature of point cloud data inhibits such measurements, and the focus of this effort was on the accuracy of the predicted convex hulls.

#### 3.1.4. Results and Discussion

Table 2 and Table 3 show the comparison of predicted defect shapes for both the ARIMA and VAR models against the ground truth. The time series generated from linear, quadratic expansion models are defined as deterministic time series, as their future value can be exactly computed by a mathematical function. These mathematical functions are $y_{t}=\theta t^{0}$ for linear expansion, and $y_{t}=\theta \times d^{\left(t-1\right)}$ for quadratic expansion (θ and d are both constants). As expected, the predicted feature state from both models completely matched with the ground truth, regardless of the defect shape. The ability to forecast such a simple deterministic process is inherent to both ARIMA and VAR modeling approaches and was used to validate basic model performance.

More interesting results can be seen for the Gaussian and uniform random stochastic time series, which are more realistic representations of defect life-cycle dynamics in practical problems. Such stochastic processes are more challenging for any predictive model. As can be seen, ARIMA models provide relatively better prediction, although the results show many similarities. The reason for the difference in predictive accuracy is the capability of ARIMA in handling nonstationary time series, as well as the assumptions of variable dependencies in VAR. Also, results show that both models can predict a defect with a Gaussian underlying process better than those with a uniform random evolution. The reason lies in the difference between the statistical properties of the two processes. Gaussian processes have a single most likely value in the distribution (the mean), whereas, in uniform distributions, every allowable value is equally likely, degrading predictive capabilities. Overall, the results of these synthetic experiments indicated that the convex hull parameterization approach and time-series modeling provides reliable and accurate representations of defect evolution across a range of defect topologies and is reasonably robust to noisy measurements. As anticipated, ARIMA provided higher prediction accuracy as stationarity assumptions became increasingly unrealistic.

### 3.2. Experimental Dataset

To further evaluate the methodology under more realistic conditions, a dataset from prior experimental testing was repurposed. In these laboratory tests, aluminum tensile coupons were tested to observe fatigue crack growth under cyclic fatigue loading. Marine-grade aluminum 5052-H32 with a nominal thickness of 2.29 mm was used. The specimen had a machined elliptical flaw in the center, and increasing load caused initiation and growth of cracks on both the right and left sides of this notch. Cycling tension loading was performed over 80,000 cycles, and the state of crack growth was captured at 30 intermediate intervals during the test, using an inspection microscope connected to a digital camera. The captured images were then segmented to isolate the crack, and the crack patterns were transformed from pixels into point clouds through binarization and spatial point sampling (Figure 8). This resulted in 2D point clouds with between 6000 and 12,000 points, depending on the size of the crack.

The convex hulls of these point clouds were computed, and feature evolutions were exported as time series, as per the methodology delineated in Section 2. An analysis of the datasets yielded an average *p*-value of 0.35, indicating that the statistical uncertainty of the experimental measurements was non-stationary. Three different tests are presented in this section to evaluate the performance of the proposed methodology including single-step prediction, multiple-step prediction, and prediction during nonlinear system dynamical behavior.

#### 3.2.1. Single-Step Prediction

Performance of the proposed algorithm for predicting a single future step is evaluated in this section. Both ARIMA and VAR model were used to find the pattern of crack growth and predict the future state of crack. The convex hulls of the right and left cracks at a load of 80,000 cycles were computed and held out as the ground truth for one single-step prediction. Results are shown in Table 4 and Figure 9. Since the performance of the ARIMA and VAR model was almost identical, the predicted shape from both models had overlap, and only the ARIMA model can be seen in the Figure 9.

#### 3.2.2. Multiple Step Prediction

To evaluate the capability of the proposed algorithm for prediction of multiple steps, 20 steps of the right-side crack, corresponding to approximately 40,000 loading intervals, were used to fit to the ARIMA and VAR models based on the HQIC criterion (Section 2.3.1). The true convex hulls of the crack at time steps 21–30 were then computed and held out as the ground truth. Then, prediction of 1–10 time steps into the future were computed and evaluated. Table 5 shows the results from the ARIMA model. Since the performance of ARIMA and VAR was very similar for this specific problem, only ARIMA is shown here. What these results reflect is, in part, the sensitivity of the time-series modeling process to the number of time-series data points or lags, used in predictive computation. For the results shown using 20 lags, predictive accuracy begins to degrade at future time steps approximately 25% the length of the total lag, in this case, five steps. Similar results were observed for models with varying numbers of lags. When fewer than 10 lags were used to fit the model, predictive accuracy was deemed unacceptable.

#### 3.2.3. Prediction during Nonlinear System Behavior

The goal of this study was to evaluate the performance of the model during a geometric nonlinearity in the evolution of a defect over time. For the crack fatigue problem studied here, there was a sudden change in the direction of crack growth after 48,000 load cycles. Of course, the predictive time-series models could not accurately forecast the convex hull immediately after this event. Rather, the question here was how long it would take the time-series models to correct for this nonlinearity in the dynamic evolution. The results for both ARIMA and VAR models are shown in Table 6 and Table 7. As can be seen, the ARIMA model quickly adapted after only two time steps (at 50,000 load cycles). The VAR model struggled to adjust for far longer, only regaining consistent predictive accuracy after 60,000 load cycles, equivalent to an additional four model time steps.

#### 3.2.4. Results and Discussion

Table 4 shows the comparison of predicted crack shape from the ARIMA and VAR models against the ground truth for cracks at the left- and right-hand sides of the notch. Figure 7 shows the predicted hull shape from both models against ground truth. Results reflect that ARIMA and VAR models both provide an accurate prediction of the future state of the crack, and the presented method is able to match the convex hull of the ground truth with high accuracy. Also, the results shown in Table 5 suggest that ARIMA and VAR models are able to predict hull shape of the crack even 10 times steps ahead with reasonable, although slightly degraded, accuracy. However, the approach is not able to mimic the true shape of the crack. Unlike the analysis performed on synthetic point clouds in the prior section, the crack shape cannot be reasonably defined by a convex polygon and, therefore, convex hull parametrization cannot represent the true shape of this polygon. However, it does predict the extremis of the visible crack tip. The evaluation of the nonlinear system dynamic prediction (Table 6 and Table 7) shows that the ARIMA model has much better performance in adapting to a nonlinear change in defect evolution and suggests that ARIMA approaches are more robust compared to VAR methods.

#### 3.2.5. Limitations of the Method

While the developed approach was shown to be effective under the experimental conditions described here, it is important to recognize the limitations of this approach. These tests were performed under controlled laboratory conditions and were not subject to the distortions and increased measurement uncertainty that arise due to environmental variations. How this measurement approach performs under unpredictable, and likely nonlinear, loading and thermal conditions remains an unstudied problem, but such conditions will undoubtedly have a negative impact on predictive performance. Field conditions are likely to degrade both the quality of generated point clouds and the predictive accuracy of any autoregressive tracking method. Furthermore, more complex material behavior, for example, highly random cracking in heterogeneous materials such as concrete, will degrade the accuracy of the autoregressive model. In general, increases in the stochasticity and nonlinearity of the underlying degradation process will result in a significant reduction in algorithm accuracy.

## 4. Conclusions and Future Work

In this work, a methodology to parametrize and model the dynamics of defect evolution based on convex hull parametrization and time-series modeling was introduced. Using convex hull parametrization, 2D synthetic and experimental point clouds representing various defect shapes and stochastic evolutions were parametrized, and their evolutions were modeled using time-series forecasting models. The future state of defects was then forecasted and evaluated against ground truth. The results indicate that this convex hull approach provides consistent and accurate representations of defect evolution across a range of defect topologies and is reasonably robust to noisy measurements; however, the behavior of the underlying dynamical process plays a significant role in predictive accuracy. Predictive accuracy degrades for both ARIMA and VAR models as defect evolution becomes increasingly nonlinear, although ARIMA is slightly more robust under such conditions.

The proposed methodology has a number of advantages over current practices. Firstly, it provides engineers with an intuitive and consistent representation of remotely sensed information over a structure’s life cycle through the reduced-dimension convex hull representation. Tracking the evolution of damages and their connections to structural performance also results in more reliable forecasting capabilities and a more complete understanding of structural performance, particularly compared to existing NDE techniques that often do not quantify damage evolution through time. This process also does not require extrapolation from other datasets for prediction; rather, it builds up a time-series representation based solely on the observed evolution of a given defect.

This study was part of an ongoing research program, and various parts of the presented methodology are being considered for further improvement. The limitations discussed in Section 3.2.3 highlight potential avenues for future work. The behavior of the algorithm under higher degrees of statistical uncertainty and material variability should be investigated. More datasets from other crack scenarios should also be considered, for instance, concrete cracking in civil infrastructure. Such studies may provide insight into how particular algorithmic aspects, such as the nearest neighbor matching aspects of hull tracking (Section 3.2), behave under complex material phenomena such as crazing or alkali–silica reactions in concrete. In such cases, the cracks may branch and split, creating unforeseen modeling challenges.

The parametrizations and hull modeling are being studied for temporal tracking of non-geometric changes such as color change in structures. The hull parametrization method is also being extended to high-dimensional feature space analyses, supporting the fusion of multiple sensors and survey information for holistic life-cycle modeling. As was presented in the methodology description, the results of this work will ultimately be used to support finite element model updating for predictive analysis of structural capacity. One notable avenue for future work is to adapt the algorithm to more realistically parameterize defect shapes using a combination of a convex and concave hull algorithm [68]. Such an approach would allow for more accurate depiction of complex geometric topologies similar to the fatigue cracks evaluated in this work. In addition, nonlinear time-series modeling methods such as recurrent neural networks may be studied for more complex defect evolutions; however, such machine-learning-driven approaches need much larger datasets to be employed.

## Figures and Tables

**Figure 1 sensors-19-04571-f001:**
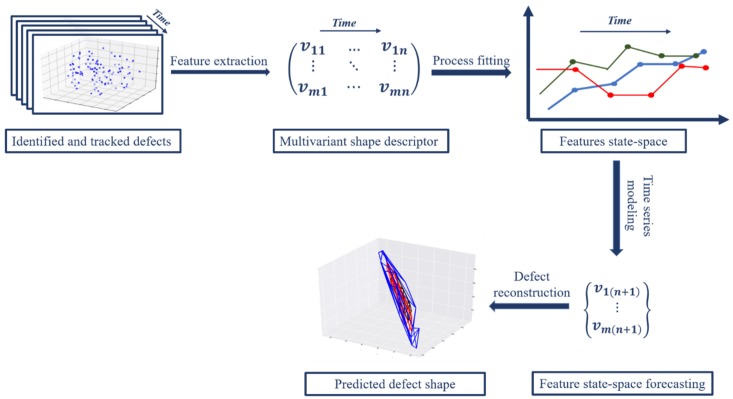
Schematic overview of the proposed methodology for life-cycle modeling of remotely sensed defects.

**Figure 2 sensors-19-04571-f002:**
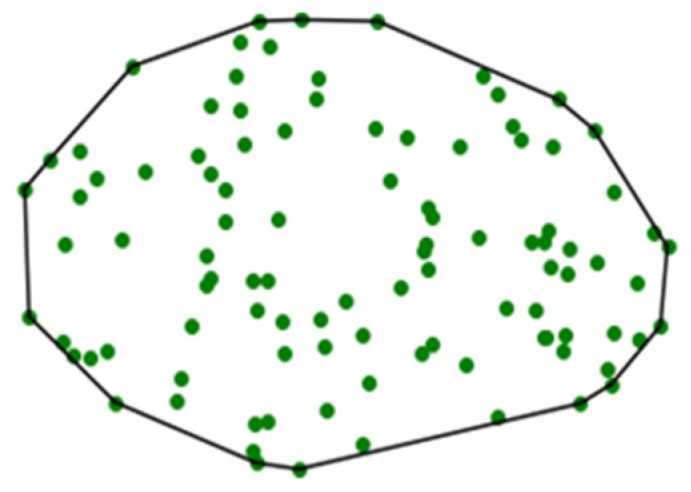
Convex hull of a point set in *R*^2^.

**Figure 3 sensors-19-04571-f003:**
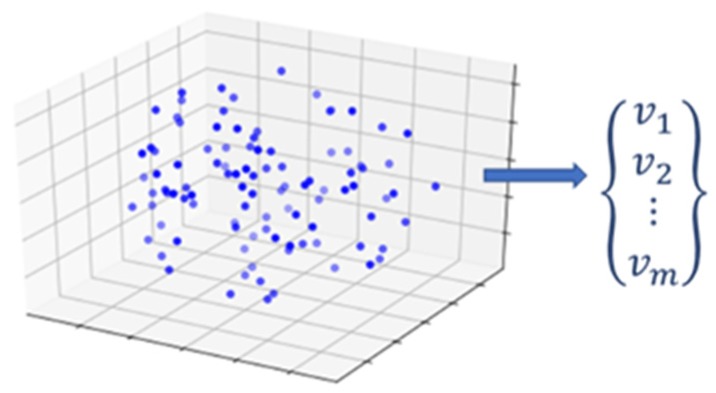
Feature extraction of convex hull vertices from a three-dimensional (3D) point cloud.

**Figure 4 sensors-19-04571-f004:**
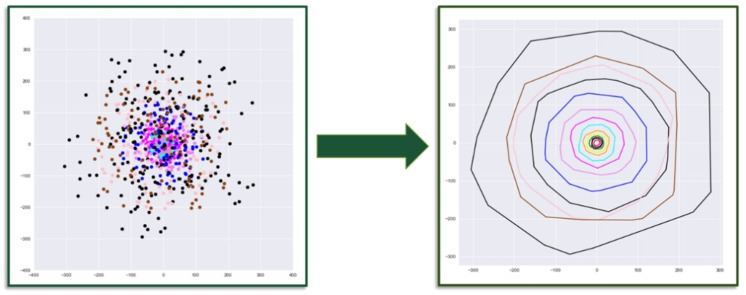
Aligning and registering hulls/clouds into a common spatial reference frame.

**Figure 5 sensors-19-04571-f005:**
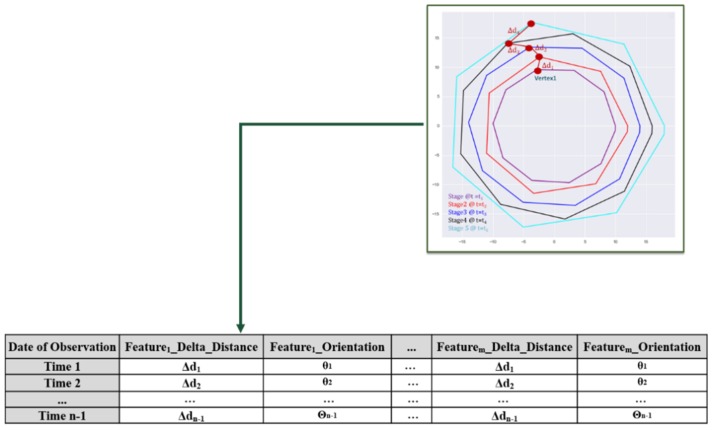
Dataset representing the extracted vertices for a time-series evolution of an arbitrary polygonal defect.

**Figure 6 sensors-19-04571-f006:**
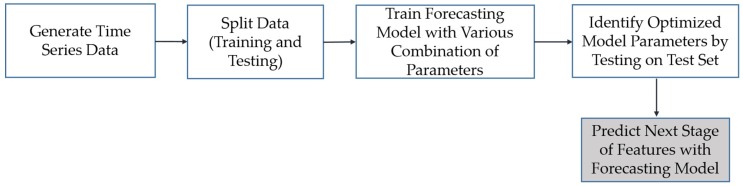
Overall time-series modeling methodology.

**Figure 7 sensors-19-04571-f007:**
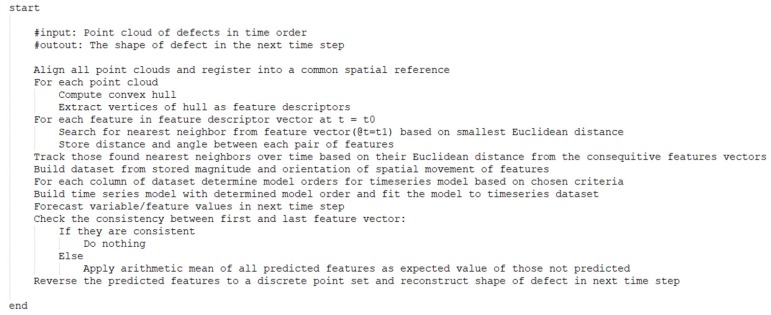
Pseudocode for the proposed methodology.

**Figure 8 sensors-19-04571-f008:**
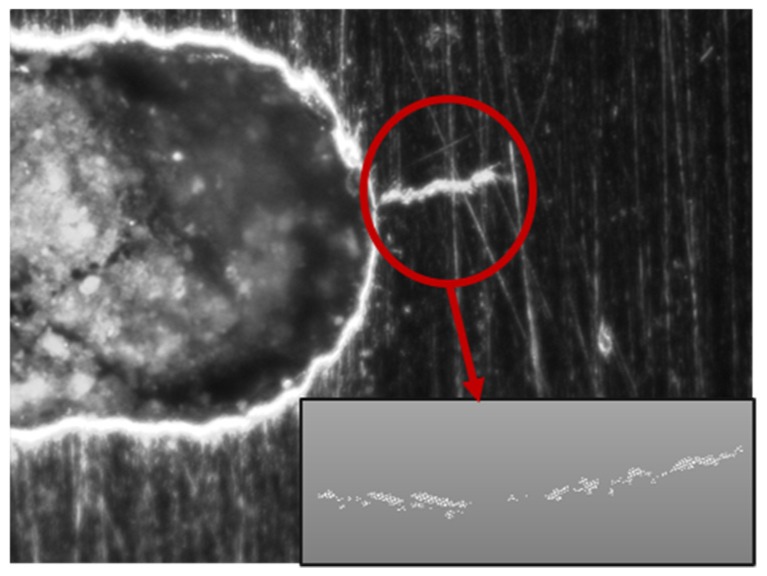
Extracted point cloud from the captured image.

**Figure 9 sensors-19-04571-f009:**
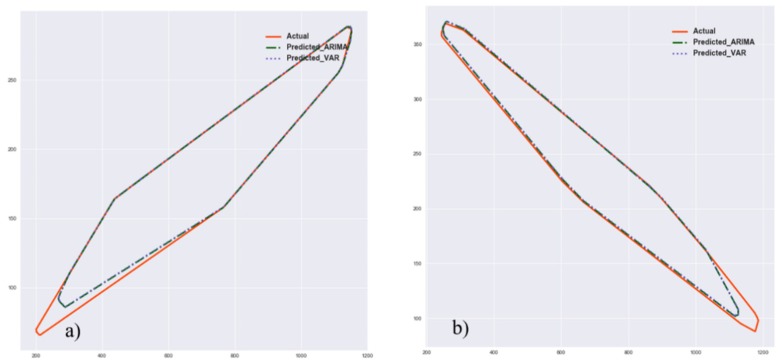
Comparison of the predicted crack shape against the ground truth for (**a**) right and (**b**) left cracks.

**Table 1 sensors-19-04571-t001:** The *p*-values from augmented Dickey–Fuller test stationary test.

Defect Shape		Triangle	Rectangle	Circle	Polygon
Defect Evolution					
**Uniform**	Linear	0.002	0.002	0.020	0.030
	Quadratic	1.000	1.000	1.000	1.000
	Random Uniform	0.950	0.950	0.390	0.940
	Random Gauss	0.960	0.960	0.990	0.950
**Non-Uniform**	Random Uniform	0.950	0.950	0.96	0.990

**Table 2 sensors-19-04571-t002:** Comparison of predicted defect shape using autoregressive integrated moving average (ARIMA) model against ground truth.

Defect Shape		Triangle		Rectangle		Circle		Polygon	
Defect Evolution									
Metrics		Overlap (%)	Area_Diff (%)	Overlap (%)	Area_Diff (%)	Overlap (%)	Area_Diff (%)	Overlap (%)	Area_Diff (%)
**Uniform**	Linear	100	0	100	0	100	0	100	0
	Quadratic	100	0	100	0	100	0	100	0
	Random Uniform	100	15	100	10	100	19	89	15
	Random Gauss	100	4	100	5	100	8	92	11
**Non-Uniform**	Random Uniform	95	7	96	5	98	4.5	87	17

**Table 3 sensors-19-04571-t003:** Comparison of predicted defect shape using VAR model against ground truth.

Defect Shape		Triangle		Rectangle		Circle		Polygon	
Defect Evolution									
Metrics		Overlap (%)	Area_Diff (%)	Overlap (%)	Area_Diff (%)	Overlap (%)	Area_Diff (%)	Overlap (%)	Area_Diff (%)
**Uniform**	Linear	100	0	100	0	100	0	100	0
	Quadratic	100	0	100	0	100	0	100	0
	Random Uniform	100	16	100	19	100	19	87	16.5
	Random Gauss	100	6	100	7.5	100	9	91	19
**Non-Uniform**	Random Uniform	92	8	94	21	95	9	85	25

**Table 4 sensors-19-04571-t004:** Comparison of predicted crack shape from ARIMA and vectorized autoregression (VAR) against ground truth.

	ARIMA		VAR	
Metric	Overlap (%)	Area_Diff (%)	Overlap (%)	Area_Diff (%)
Right Crack	100.0	7.0	100.0	7.0
Left Crack	99.0	5.0	96.0	5.0

**Table 5 sensors-19-04571-t005:** Comparison of predicted crack shape from ARIMA against ground truth.

	**1 Step**	**2 Steps**	**3 Steps**	**4 Steps**	**5 Steps**
Overlap (%)	100	100	99	98	96
Area_Diff (%)	1	1	4	3	4
	**6 Steps**	**7 Steps**	**8 Steps**	**9 Steps**	**10 Steps**
Overlap (%)	95	94	93	92	91
Area_Diff (%)	3	6	5	3	3

**Table 6 sensors-19-04571-t006:** Comparison of the predicted crack shape from ARIMA model against ground truth.

Load	48,000	50,000	52,000	54,000	58,000
Overlap (%)	83	96	95	99	95
Area_Diff (%)	8	14	17	15	3

**Table 7 sensors-19-04571-t007:** Comparison of predicted crack shape from VAR model against ground truth.

Load	48,000	50,000	52,000	54,000	58,000	60,000	62,000	64,000
Overlap (%)	63	70	66	92	70	95	87	97
Area_Diff (%)	8	14	17	15	16	5	2	6

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
