# Peer review of "Life-Cycle Modeling of Structural Defects via Computational Geometry and Time-Series Forecastingâ€"

_sensors, 2019, doi:10.3390/s19204571_

Round 1
Reviewer 1 Report
This paper presents an interesting approach of detecting and forecasting the behavior of defects in the materials. It utilizes the cloud points as reference points of indicating the defect and then extract the increasing pattern of defect to understand how it grows. The idea is in fact not very new. Not too many novel algorithm or experimental approaches. However, the method could be further developed to a more mature one to consider real practices. Certain key points I would like to highlight:
The authors perhaps do not have too much real practice experiences. In engineering measurement, the defect such cracks in concrete is very difficult to capture the changes from cloud point. The changes are usually too small compared to other operational effects. Things like loading changes or temperature changes would have more influence to the defect observation done by cloud, how to filter these uncertainties is a significant question.
The transfer of geometric information to the time series information is not necessary. Why we need to downgrade the information from 2D to 1D? I believe we can get more property characteristics from 2D observation.
In the proposed flow chart, the authors indicated that the final stage is to input the information of defect into the FEM. However, this is never mentioned in the later case study neither theoritical one nor experimental one, If the authors did not do this FEM study, they should not refer it.
The limitations of this approach is missing. As I said, there are plenty uncertainties that the authors did not consider. In fact, the uncertainties of this prediction, from my point of view, should be high. E.g. in concrete materials, the material randomness is usually very high, such defect prediction can always get wrong, as the defect can come out very randomly in the materials. Other issues should also be mentioned in the study.
The work actually lacks a significant part of introducing the equipment setup. As this journal is the journal "sensors", the authors are required to give a detailed description about the experiment equipment. How did they measure the cloud points? What equipment is used to collect the information? These are all missing.
Reviewer 2 Report
The authors present a methodology for the prediction of crack growth in structures based on measurements from non-contact measurement techniques. They use several advanced signal processing and reduction techniques for the prediction of crack growth.
The paper has merit but needs some work before it can attain level suitable for publication in an journal
The major concerns are:
Literature review: The list of citations is really long (68) but not a lot of work has been discussed in detail. So some effort pointing toward the state of the art with relevant shortcomings and advantages needs to be provided
Methodology: Some work needs to be done in the description of the methodology.
Section 2.1 has been covered in detail and explained well. Unfortunately the sections 2.2 and 2.3 need to be clarified. Putting some explanatory figures flowcharts may help the clarity
Results:The design of the research is acceptable but may be improved. More data sets from other crack situation may be considered. Considering the application of the methodology is more common in civil engineering applications some cases of cracks in concrete structures may have to be provided. The reviewer is particularly curious about the nearest neighbour algorithm working in case of cracks due to crazing, alkali-silica reaction where the cracks may split.
Also a discussion on the sensitivity of the prediction to the number of time series data used needs to be provided. A quantitative and qualitative comparison may be provided.
In addition there are several typos and grammatical mistakes that need to be addressed/
Round 2
Reviewer 1 Report
The paper is well revised. I recommend its further publication in the journal.
Reviewer 2 Report
The paper covers a crack quantification and prediction method.
The paper has been significantly improved since the last round of review.
The key aspect of the paper which the other reviewer had raised and still not appropriately addressed is how the paper fits under the scope of the sensors journal. May be the authors will consider some other journal such as applied sciences which is more suitable for publication